# Cytokine Signalling at the Microglial Penta-Partite Synapse

**DOI:** 10.3390/ijms222413186

**Published:** 2021-12-07

**Authors:** Jason Abbas Aramideh, Andres Vidal-Itriago, Marco Morsch, Manuel B. Graeber

**Affiliations:** 1Brain and Mind Centre, Faculty of Medicine and Health, The University of Sydney, Sydney, NSW 2006, Australia; jason.aramideh@sydney.edu.au; 2Faculty of Medicine, Health & Human Sciences, Macquarie Medical School, Macquarie University, Sydney, NSW 2109, Australia; andres.vidal-itriago@hdr.mq.edu.au (A.V.-I.); marco.morsch@mq.edu.au (M.M.)

**Keywords:** glial inflammation, microglial cytokines, microglia-derived brain macrophages, neuroinflammation, synaptic plasticity

## Abstract

Microglial cell processes form part of a subset of synaptic contacts that have been dubbed microglial tetra-partite or quad-partite synapses. Since tetrapartite may also refer to the presence of extracellular matrix components, we propose the more precise term microglial penta-partite synapse for synapses that show a microglial cell process in close physical proximity to neuronal and astrocytic synaptic constituents. Microglial cells are now recognised as key players in central nervous system (CNS) synaptic changes. When synaptic plasticity involving microglial penta-partite synapses occurs, microglia may utilise their cytokine arsenal to facilitate the generation of new synapses, eliminate those that are not needed anymore, or modify the molecular and structural properties of the remaining synaptic contacts. In addition, microglia–synapse contacts may develop de novo under pathological conditions. Microglial penta-partite synapses have received comparatively little attention as unique sites in the CNS where microglial cells, cytokines and other factors they release have a direct influence on the connections between neurons and their function. It concerns our understanding of the penta-partite synapse where the confusion created by the term “neuroinflammation” is most counterproductive. The mere presence of activated microglia or the release of their cytokines may occur independent of inflammation, and penta-partite synapses are not usually active in a neuroimmunological sense. Clarification of these details is the main purpose of this review, specifically highlighting the relationship between microglia, synapses, and the cytokines that can be released by microglial cells in health and disease.

## 1. Glia Interact with Synapses in Health and Disease

The ability to adapt and protect neuronal circuitry throughout an animal’s lifespan is a remarkable feature of the CNS. Synaptic plasticity refers to the process by which the nervous system generates new synapses, eliminates unneeded ones and modifies the electrophysiological, molecular and structural properties of existing synapses in response to stimuli.

Synaptic plasticity generally depends on reciprocal interactions between neurons and glial cells and the extracellular matrix (ECM) in what has been described as the ‘tetrapartite synapse’ [1,2]. The concept of the synapse has evolved over the last two decades from involving pre- and post- synaptic neuronal elements to including a third element—astrocytes—resulting in a ‘tripartite’ synapse [3]. This further evolved to include a fourth component—that is, the ECM [2], which is difficult to discern in CNS parenchyma—to form the ‘tetrapartite synapse’. In this ensemble, astrocytic processes engulf pre- and post-synaptic elements and express EAAT1-EAAT2 glutamate transporters that facilitate rapid re-uptake of excess glutamate limiting excitatory synaptic stimulation. Astrocytes also release gliotransmitters such as ATP and GABA [4]. The ECM provides scaffolding, which is important for maintaining synaptic integrity. Specifically, in the context of the tetrapartite synapse, the ECM serves to retain neuronal connectivity [5], which impacts the diffusion and localisation of key molecules such as neurotransmitters, ions and membrane receptors, and it also contains matrix molecules such as chondroitin sulfate proteoglycans (CSPGs), which inhibit neurite and axon growth [6]. In addition to the growing literature on the role of the ECM, astrocytes and NG2 cells [7] on synaptic plasticity, microglial cells are now also considered essential players. De Leo, Tawfik and LaCroix-Fralish [1] deserve credit for drawing attention to the presence of the tips of microglial cell processes at a subset of CNS synapses and coining the term tetrapartite synapse originally.

Microglia and brain macrophages, which can be derived from microglia as well as other sources, are traditionally understood to play a role in innate immune responses. This has resulted in associating microglial activity with immunological reactions by default and microglial relations with synapses to be referred to as ‘neuroimmunological’ [8]. This is incorrect. Microglia in healthy brain have been shown to interact with synapses in the absence of any inflammation as well as in pathological conditions [9]. Some of the earliest evidence for microglia-derived cytokines regulating synaptic function pin-pointed TNF-α as a regulator of the amount of AMPA receptors and synaptic strength in the hippocampus [10,11,12]. Transcriptomic analysis revealed that microglia rather than astrocytes produce TNF-α transcripts [13]. Fractalkine, interleukins, interferons and growth factors such as brain-derived neurotrophic factor (BDNF) derived from microglia have also been shown to have a role in synaptic plasticity [14,15,16,17]. Abnormally functioning microglia displaying dysregulated cytokine levels have been associated with abnormal synaptic structure and function in neurodegenerative disease models of Alzheimer’s Disease (AD) and autism (ASD) [18,19,20]. More recently, abnormal cellular bioenergetic states such as caloric restriction and ketogenic diet as well as the glycolytic inhibitor 2-deoxyglucose (2DG) have been demonstrated to influence microglial transcription of cytokine genes [21,22]. York et al. [23] reported metabolic reprogramming of microglia stimulated by lipopolysaccharide (LPS), which resulted in glycolysis necessary for IL-1β production. This process could be reversed by the addition of 2DG, which resulted in the inhibition of HIF-1α accumulation and IL-1β production, and rescued long-term potentiation (LTP) of synaptic plasticity. Overall, microglia and their cytokines are now recognised to play an indispensable role in plasticity of the tetra-partite synapse.

Since tetrapartite may also refer to the presence of extracellular matrix components, we propose the more precise term microglial penta-partite synapse for synaptic contacts that show a microglial cell process in close physical proximity to neuronal and astrocytic synaptic elements. Thus, microglial penta-partite synapses comprise a smaller subset of synapses. Precise topoanatomical data on which neuronal circuits if any are preferentially involved are currently lacking. We also do not know whether microglial cell processes have a clear affinity for specific types of synapses and whether they contact the same synapses again and again, i.e., how transient any topoanatomical penta-partite synapse maps are.

Furthermore, we discuss the controversial term ‘neuroinflammation’. AD, Parkinson’s Disease (PD) and schizophrenia (SCZ) are often conceptually linked to ‘neuroinflammation’. Yet, typical inflammation pathology is missing in these diseases, and molecular genetic evidence in fact shows that they are categorically non-inflammatory conditions [24]. We propose instead that dysfunction of the microglial penta-partite synapse is of key relevance to CNS disorders that feature microglial activation.

## 2. Microglial Activation Is Not (*Neuro**) Inflammation*

### 2.1. What Is Inflammation?

Inflammation is a multicellular, multistage and dynamic response to tissue damage, characterised by the infiltration of cells of the adaptive immune system [25,26]. The CNS is no exception to this relatively common pathology, which can affect all tissues of the body. The specific tissue characteristics of inflammation are very well established and the exclusion or confirmation of their presence forms part of the routine diagnostic microscopic algorithm in histopathology. Classical examples of inflammation in the CNS include bacterial or viral infections, autoimmune conditions such as multiple sclerosis but also ischaemia/stroke.

A number of authors refer to inflammation as “neuroinflammation” when it occurs in the nervous system. In other words, “neuro” refers to topology only in this case and is strictly speaking tautological. Other authors, however, apply the term “neuroinflammation” to entirely different pathological processes that are specific to the nervous system, lack participation of peripheral immune cells and display microglial activation. Milatovic et al. [27] defined “neuroinflammation” as the activation of microglia and astrocytes accompanied by the upregulation of several pro-inflammatory cytokines, such as interleukin (IL)-1β, IL-6, and tumour necrosis factor (TNF-α), and chemokines such as CCL2, CCL5, CXCL1, secondary messengers (NO and prostaglandins) and reactive oxygen species [28,29]. However, this is not sufficient for a diagnosis of inflammation. Examples of cases in the literature where the term “neuroinflammation” has been extended in such a way include Alzheimer’s Disease (AD), Parkinson’s Disease (PD), schizophrenia (SZ), as well as etiologically highly diverse conditions such as obesity, autism, major depression, bipolar disorder, anxiety, air pollution and epilepsy amongst others. Neither infiltrating innate nor adaptive immune cells are typically found in Alzheimer’s Disease (AD), Parkinson’s Disease (PD) and schizophrenia (SZ) to any significant extent. Recently, Gate et al. [30] suggested a role for the adaptive immune system in blood and cerebrospinal fluid in AD based on a small number of clonal, antigen-experienced specialised CD8^+^ T effector memory CD45RA^+^ (T_EMRA_) cells patrolling intrathecal cerebrospinal fluid (CSF), as well as CD8^+^ T cells in the hippocampus near beta-amyloid plaques. In general, neither AD, PD nor SZ demonstrate perivascular mononuclear cell infiltrates either, but they have been assigned to the category of ‘neuroinflammation’ by some authors on the basis of observed microglial activation, particularly with reference to translocator protein positron emission tomography (TSPO PET) imaging results [22]. There is no generally accepted neuropathological tissue correlate of ‘neuroinflammation’ [24], and many if not most publications that use the term do not define it.

Applying the same scientific term, inflammation, to categorically different disease entities represents a logical impossibility. The term “neuroinflammation” is therefore not viable. Unfortunately, as long as the confusion persists, scientific progress will be inhibited. Most relevant in our view, the ambiguity of “neuroinflammation” inhibits research interest in what happens at the microglial penta-partite synapse.

### 2.2. Microglia Are Never-Resting Sensors of Pathology

Microglia are considered activated when there is, at a minimum, increased expression of markers that are absent from microglia in healthy CNS tissue. This was first described for certain lectins in 1987 [31] and complement receptors in 1988 [32] but may also include the production of inflammatory mediators such as cytokines [33]. As discussed by Cherry et al. [34], concepts concerning microglial phenotype polarisation have been carried over from literature on peripheral macrophages throughout the last century. One particular such concept that is of concern is that microglia in the healthy CNS exist in a resting state [35]. This concept has been translated into the idea that cytokine expression by microglia is reflective of activation. This is problematic primarily because (1) microglia, as we know today, should never be considered “resting”—they are constantly “surveying” the microenvironment—and (2) cytokine expression occurs in non-activated microglia as well.

Some studies have explored the role of microglial activation in ALS [36,37], AD [38,39] and MS [40,41,42]. While these studies report significantly increased microglial activation, as inferred from increased TSPO binding, some studies have demonstrated no correlation between microglial activation and increased TSPO binding [40,41,42]. Owen et al. [43] reported that TSPO expression did not increase with classical pro-inflammatory activation in primary human microglia, unlike in rodent microglia. Hafizi et al. [44] reported no significant differences between patients and healthy volunteers with regard to microglial activation using [^18^F] FEPPA in either the dorsolateral prefrontal cortex or hippocampus. Nutma et al. [45] found that although TSPO expression may provide a general marker for glial activation in multiple sclerosis, such interpretations depended on pathological context. As with AD [46], increased TSPO PET signal may reflect differences in microglia and astrocyte abundance [45]. Increased TSPO expression was not associated with solely pro- or anti- inflammatory microglia phenotypes in MS, but rather with an intermediate activation state demonstrated through the expression of both CD40 and CD206 [45]. Sneeboer et al. [47] found no positive correlation between TSPO expression and microglial activation markers in either patients or controls for schizophrenia. Microglia constantly require ATP as an energy source for their baseline dynamics and maintenance of cell morphology [48]. A positive correlation between TSPO and increased ATP production suggest that TSPO expression reflects an increased demand for energy [49]. This had been previously demonstrated by Banati et al. [50], who reported TSPO^−/−^ microglia producing significantly less ATP. Taken together, the findings suggest that interpreting TSPO imaging as a reflection of microglial activation needs to be treated with caution, particularly given that increased ATP production in microglia does not necessarily suggest inflammation-induced activation.

Molecules associated with immune functions in peripheral organs such as certain cytokines can have additional CNS-specific roles independent of their conventional roles in generating an inflammatory and/or immune response. A number of such molecules, IL-1β, IL-γ, CCL2, major histocompatibility complex (MHC) proteins, DAP-12, and the CR3 complement receptors have been implicated in synaptic plasticity [51]. The role of some major cytokines in microglial-mediated synaptic plasticity has been reviewed by Werneburg et al. [52] and thus will not be discussed further here.

### 2.3. Cytokines Are Present in Unstimulated Microglia

Cytokine levels in microglia are typically increased in response to pathological stimuli [53]. For instance, both CNS neurons and glial cells express low levels of interleukin-6 (IL-6) and IL-6R in physiological conditions [54]. In contrast, under pathological conditions, such as following peripheral nerve injury, cytokines and their receptors, such as IL-6/IL-6R, have been shown to be increased with microglial activation [55]. IL-6 utilises glycoprotein 130 (gp130) signalling via binding with IL-6R, which activates intracellular Janus kinase (JAK) tyrosine-kinase, which then activates the signal transducer and activator of transcription (STAT) [56]. Figure 1 illustrates this process in the context of miR-21-5p mediated neuroprotection following traumatic brain injury [56,57].

Microglia release cytokines that can have neuroprotective roles without stimulation [58]. Studies have identified cytokine expression in unstimulated microglia [59,60,61]. In the absence of recognised stimulatory signals, microglia are the primary source and secretors of cytokines in the CNS. Lee, Nagai and Kim [59] reported mRNA transcripts for interleukin-1β (IL-1β) -6, -8, -10, -12, -15, tumor necrosis factor-α (TNF-α), macrophage inflammatory protein-1α (MIP-1α), MIP-1β, and monocyte chemoattractant protein-1 (MCP-1) in unstimulated human microglia.

Elkabes, DiCicco-Bloom and Black [61] demonstrated neurotrophins NGF, BDNF, NT-3 and NT-4 in microglia regulating microglial proliferation. Other studies have demonstrated the expression of several cytokines in different CNS cell types in both humans and animal models under physiological conditions, including retina Muller glial cells [62], cerebellar neurons and glia [63], as well as microglia obtained from human brain tissues [64,65]. Under non-stimulated conditions, human microglia express various interleukin family cytokines, TNF-α, macrophage inflammatory protein (MIP)-1α, MIP-1β and MCP-1, as well as receptors for some of these cytokines, as shown in Table 1 below [66]. CXC chemokines such as fractalkine (CX3CL1/CX3CR1), SDF-1α, and CXCL12/CXCR4/CXCR7 have been reported in the healthy CNS of mice and human microglia [67,68] and neurons [69,70,71,72,73]. Similarly, β-chemokines (or CC chemokines) such as MCP-1 (CCL2), MIP-1α (CCL3), MIP-1β (CCL4), RANTES (CCL5) and C10/MRP-1 (CCL6) have also been identified in both healthy neurons [69,70,71,72,73] and microglia [74].

Several studies have reported MIP-1α (CCL3), MIP-1β (CCL4) and RANTES (CCL5) mRNA expression across several brain regions in healthy animals, such as the cerebral cortex [75], thalamus, striatum, hippocampus, midbrain [76,77], ventromedial hypothalamus [78] and cerebellum [79]. Ciechanowska et al. [76], however, observed no statistically significant changes in CCL3 protein expression levels when compared to the thalamus of traumatic brain injury (TBI) mice. Several studies have also reported microglia as the primary source of CCL3 and CCL6, as well as neurons [80]. CCR3 has also been demonstrated in both neurons and microglia of several CNS regions, such as the hippocampus, cerebral cortex, brain stem, cerebellum and spinal cord [81,82,83,84]. Lanfranco et al. [77] identified CCL5 mRNA expression by glia cells, but not by neurons, in the corpus callosum. CCL5mRNA was detected more predominantly in neurons than in glial cells of the cerebral cortex, hippocampus and midbrain [77]. In summary, these studies suggest that several cytokines associated with inflammation in peripheral organs such as fractalkine (CX3CL1/CX3CR1), SDF-1α, CXCL12/CXCR4/CXCR7, MCP-1 (CCL2), MIP-1α (CCL3), MIP-1β (CCL4), RANTES (CCL5) and C10/MRP-1 (CCL6) are expressed in the normal CNS as well as during the inflammatory response.

In recent times, microglia have been shown to reprogram their cellular metabolism to accommodate functions needing cytoskeletal changes, functional remodelling and cytokine synthesis [85,86,87]. Murine brain microglia express all genes required for glycolytic and oxidative energy metabolism, including the expression of transporters for all energy substrates [13,88,89]. In the absence of glucose such as during anti-inflammatory states, microglia utilise oxidative phosphorylation by metabolising alternate carbon sources such as glutamine [90], and fatty acids from lipid droplets (LDs) in glucose-deprived microglial cells [91,92]. During pro-inflammatory conditions, microglia use aerobic glycolysis for energy production. Glucose fluctuations activate signalling pathways such as MAPKs and PI3K/Akt, which form the stress that alters NF-κB activity in microglia [93]. Yamada et al. [94] demonstrated using real-time RT-PCR that miR-16-5p, miR-196b-5p, and miR-218-5p expression levels were significantly higher in caloric-restricted mice compared with controls. Specifically, miR-16-5p was found to suppress mRNA expression of IL-1β, IL-6, and TNF-α, further suggesting that caloric restriction suppresses cytokines through transcriptional control [94]. Bernhart et al. [95] proposed that upregulation of glycolytic enzymes in response to lysophosphatidic acid (LPA) treatment of microglia could serve two purposes: (i) generating a survival signal by generating sufficient amounts of ATP and (ii) a source of cellular energy to fuel plasma membrane located, ATP-dependent ion pumps. Wang et al. [96] found that GLUT1 was significantly increased to promote anaerobic glycolysis following LPS  +  IFNγ stimulation, and that blocking GLUT1 reduced anaerobic glycolysis, reprogrammed microglial metabolism to aerobic glycolysis and suppressed phagocytosis in microglia. These findings suggest that microglial mitochondrial reprogramming control cytokine-mediated microglial functions.

Taken together, the increased appreciation of the role of microglial cytokines in physiological conditions independently necessitates the need to reconsider usage of the term “neuroinflammation”. The “neuroinflammatory” paradigm wrongly disregards the active role microglia play in physiological functions.

## 3. The Microglial Penta-Partite Synapse

TSPO PET imaging and histopathological studies of diseased brains have revealed microglial activation across a wide range of brain pathologies. What we do not know with certainty yet, although this seems likely, is whether the microglia activated in these conditions all have cell processes that are in direct contact with synapses, at least temporarily. Microglial cells are known to “check” on synaptic integrity in regular time intervals [48,97], reaching some synapses. Similarly, astrocyte processes are not present at all synapses (cf. below). Figure 2 provides a schematic drawing of a microglial penta-partite synapse.

### 3.1. Neuron-Glia Bidirectional Communication Is Mediated by Purinergic Signalling

Astrocyte–neuron bidirectional communication mediated by purinergic signalling has been well documented by in vivo as well as ex vivo studies [98]. There is substantial evidence suggesting that increased neuronal activity, as evidenced by neurotransmitter release, results in increased GPCR-mediated intracellular astrocytic Ca^2+^ signalling and astrocyte glutamate release [99,100,101].

Purinergic receptors and purines have been implicated in neuron–glia interactions during various physiological and pathological conditions. Neuronal release of ATP has been shown to directly act on astrocytic purinergic receptors as either ATP or as a breakdown product, such as ADP or adenosine [102]. Juaristi et al. [103] found that astrocytes upregulate glycolysis, lactate and pyruvate production in mitochondria in response to intracellular Ca^2+^ increases following ATP and glutamate stimulation of P2YR. Shigetomi et al. [104] reported that increased P2Y1R expression triggered increased intracellular Ca^2+^ in astrocytes. Recently, Monai et al. [105] reported astrocytic GPCR activation as the prevalent mechanism of transcranial direct current stimulation (tDCS)-induced Ca^2+^ surges. Muller and Taylor [106] found that ATP-evoked Ca^2+^ signals in cultured human fetal astrocytes are mediated by P2Y1 and P2Y2 receptors.

Microglial communication with both neurons and astrocytes is well documented. For instance, microglia are known to alter excitatory neurotransmission through an ATP-dependent astrocyte communication mechanism [107]. Microglia are also known to be stimulated by neuronal activity-dependent mechanisms to eliminate excessive synapses during neurodevelopment [108]. Purinergic receptors found on microglia have been demonstrated to mediate microglial communication with astrocytes and neurons. Janks et al. [109] found that cultured human microglia express P2X7Rs that stimulate ATP-gated cationic currents. Beneventano et al. [110] reported rapid microvesicle shedding from BV2 microglia following P2X7R activation. Liu et al. [111] showed that P2X7R inhibition reduced IL-1β expression, P38 phosphorylation, and glial activation in the cerebral cortex and improved neurobehavioural outcomes following traumatic brain injury. Several purinergic receptor subtypes have also been identified in microglia, specifically: P2Y1, P2Y2, P2Y2/4, P2Y6, and P2Y12, with P2Y6 and P2Y12 receptors appearing to be functionally important [112]. Several P2YRs have also been identified in astrocytes, such as the above and other subtypes: P2Y11, P2Y13, and P2Y14 receptors [112]. Pascual, Achour, Rostaing, Triller and Bessis [107] found that in vitro microglia enhance excitatory postsynaptic neurotransmission by releasing ATP which stimulates astrocyte P2Y1R. Jiang et al. [113] reported both ATP and ADP mediated mechanical stimulation-induced intercellular Ca^2+^ wave communication via P2Y_12/13_ receptors in BV-2 microglia. Quintas et al. [114] found that the microglial P2YR inhibition of astrocyte proliferation occurs via the release of IL-1β, IL-1α and TNF-α. Furthermore, the absence of P2Y12 has been shown to lead to impaired microglia process motility following injury [48]. Microglial expression of P2Y_6_ receptors has also been shown to trigger microglial phagocytosis following hippocampal excitotoxicity [115]. Collectively, these findings suggest that purinergic receptors may play an important role at the microglia penta-partite synapse. Table 2 provides a summary of microglial mediators and their receptors that have been shown to be involved in synaptic plasticity.

### 3.2. Astrocytes and Microglia Are Not Present at All Synapses

Although astrocytes can contribute to the processing, transfer and storage of information, they are not present at all synapses. Most studies report a ‘*non-uniform distribution*’ of perisynaptic astrocyte processes along synapses with a subset of synapses (around 40%) undergoing astrocyte-induced potentiation following post-synaptic depolarisation in the hippocampus [116]. In vivo two-photon imaging has shown that microglia processes make frequent, direct and transient contact with synapses in the uninjured CNS [97]. Akiyoshi et al. [117] further characterised the microglia–synapse contact as resulting in increased neuronal activity in vivo with awake mice. Tremblay, Lowery and Majewska [9] further described this relationship and demonstrated protrusions of microglial processes apposing dendritic spines, presynaptic terminals, the synaptic cleft and perisynaptic astrocyte processes.

Augusto-Oliveira et al. [118] suggest that astrocyte process contact with neuronal synapses ranges from 50 to 60% of all synaptic structures, depending on CNS region. Hippocampal synapses have demonstrated astroglial coverage ranging from 60 to 90%; specifically, Witcher et al. [119] found astrocyte processes in 40–60% of thin, mushroom spine-axon synapses in the hippocampus, while 60–99% of larger mushroom spines and spines with perforated synapses were found to be in contact with astrocyte processes. Ventura and Harris [120] demonstrated that 57 ± 11% of the hippocampal excitatory synapses in mature rats had astrocytic processes apposed to them. Astrocytic processes in this synapse subset surrounded less than half of the entire synaptic interface. In contrast, the number of synaptic spines contacted by microglia is significantly lower. Weinhard et al. [121] found that only 3% of spines were in contact with microglia, taken from a total of 8900 spines from secondary dendrites of GFP+  neurons in the CA1 stratum radiatum of fixed hippocampal tissue at P15. About 1.9% of spines contacted by microglia presented with minor microglia-spine co-localisation [121].

In the cerebellum, Xu-Friedman et al. [122] found that glial processes cover approximately 90% of climbing fibre synapses and approximately 65% of parallel fibre synapses on Purkinje neurons. Spacek [123] reported that 74% of cerebellar Purkinje cell synapses were ensheathed by astrocytes as opposed to 29% of dendritic spines in the mouse visual cortex. Broadhead et al. [124] identified glial presence in around 56% of synapses in the mouse spinal cord. Bernardinelli et al. [125] reported astrocytic process contact as high as 90% in the somatosensory cortex layer IV, particularly with the post-synaptic element [126]. Collectively, these observations suggest that synapses featuring astrocytes represent a subset. The subset of penta-partite microglial synapses is less frequent. In addition, the latter may be significantly modulated by microglial activation.

Since glial cell processes are not present at all synapses, the question arises under what circumstances can we expect their presence? Astrocytes have been shown to discriminate between the activity of different synapses belonging to different axons, suggesting synapse-specificity among astrocytes. Earlier work by Perea and Araque [127] found that rises in Ca^2+^ in astrocytes can be evoked by acetylcholine and glutamate released from Schaffer collaterals, but not glutamate released by alveus stimulation. Quantitative 3D analyses by Ventura and Harris [120] identified astrocytes at 88% of perforated synapses, 61% of single synapse boutons (SSBs), 52% of macular synapses, and 40% of multisynaptic boutons (MSBs). Xu-Friedman, Harris and Regehr [122] discovered differences in astrocyte ensheathment between parallel fibre synapses and climbing fibre synapses of Purkinje cells, suggesting a preference of astrocytes for climbing fibre synapses as opposed to parallel fibre synapses. Witcher, Kirov and Harris [119] found that the post-synaptic density (PSD) and mean synapse size were greater when astroglia was present at the post-synaptic dendritic spine, suggesting that larger synapses were more likely to have astrocytes than smaller synapses. With regard to microglia, Tremblay, Lowery and Majewska [9] found that microglial processes preferentially localised to small and structurally dynamic dendritic spines rather than larger dendritic spines in the primary visual cortex (V1) of juvenile mice. Another suggestion is that glia are present at active synapses where glutamate is present. This ties in with the former suggestion that astrocytes may have a preference for synapse size because local efficacy of glutamate uptake is lower at large spines [128]. Henneberger et al. [129] found that LTP triggered the withdrawal of astroglial processes from potentiated synapses, thereby boosting glutamate spillover and NMDA-receptor-mediated inter-synaptic cross-talk. Microglial contact with synaptic elements have also been shown to be modulated by sensory deprivation [97]. Umpierre et al. [130] recently found that hyperactive shifts in neuronal activity triggered increased microglial process calcium signalling and process extension in awake mice. It is likely that a higher number of penta-partite synapses exists in the living brain compared to tissue slices (cf. Conclusions and [119]).

### 3.3. Ultrastructural Analysis of Microglia Penta-Partite Synapses

Microglia cell somata can be found in juxtaposition to neuronal cell bodies, particularly in the cerebral cortex (“satellite microglia”), and their processes may also appose to dendrites, dendritic spines, pre- and post-synaptic elements and even patrol the synaptic cleft, especially under pathological conditions [131,132]. In one of the earliest ultrastructural studies on axotomised facial nuclei, Blinzinger and Kreutzberg [132] identified microglial somata and stem processes surrounding large surface areas of motorneuron cell bodies. Tremblay et al. [133] identified microglial cell bodies typically surrounded by pockets of extracellular space, with large and small processes in direct juxtaposition to synaptic elements such as dendritic spines, axon terminals and perisynaptic astrocytic processes in saline-injected mice. Savage et al. [134] similarly observed pockets of extracellular space around microglial cell bodies in both saline- and LPS- injected mice. In another study, Savage et al. [131] found that wild-type microglia from the dorsomedial striatum interacted with excitatory synapses irrespective of age. More recently, the microglia relationships with synapses have been further characterised by trogocytosis or ‘*nibbling*’ of presynaptic boutons and axons rather than by (large-scale) phagocytosis [121]. It is proposed that the microglial trogocytosis of presynaptic boutons and axons are aimed at reducing or remodelling axonal processes [121].

### 3.4. Factors Influcening the State of the Penta-Partite Synapse

Studies on experience-dependent synaptic plasticity have demonstrated that the highly dynamic remodelling of synapses occurs across the entire lifespan of an organism and not just during the neurodevelopmental stages [135,136]. The in vivo two-photon imaging of fluorescent-labelled neurons and microglia by Wake et al. [97] demonstrated neuronal activity-dependent synaptic contact by “patrolling” microglial processes at a frequency of about once per hour in mice. Altering light exposure has also been shown to induce changes in contact between microglia processes and synaptic profiles in the visual cortex of juvenile mice [9]. Studies examining the role of microglia in monocular dominance (MD) have demonstrated a shrinkage and loss of thalamocortical synapses in the mouse primary visual cortex (V1) [137,138,139]. Sipe et al. [138] demonstrated in vivo that microglia depend on the G-inhibitory-protein purinergic receptor, P2Y12R, when responding to changes in cortical activity by increasing process arborisation, reducing process motility and increasing interactions with synaptic elements. Stowell et al. [140] had shown that pharmacologic stimulation of β2-adrenergic receptors in microglia disrupted experience-dependent plasticity.

Recent studies have also explored the role of key microglial microRNAs (miRNAs) that regulate the expression of several proteins in synaptic stability and microglial function. Prada et al. [141] observed microglia-to-neuron miR-146a-5p transfer and Syt1 and Nlg1 downregulation that resulted in a significant decrease in dendritic spine density in rat hippocampal neurons in vivo and in vitro. Interestingly, Prada et al. [141] found that the administration of IL-1β, TNF-α, and IFN-γ promoted extracellular vesicle (EV) release from microglia, which delivered miR-146a-5p. Sun et al. [142] reported the downregulatory role of miR-324-5p on astrocyte CCL5 expression, which resulted in the synapse loss of mice neurons. Tsai et al. [143] reported that drosophilia larvae lacking miR-274 released by glia did not develop complete synaptic boutons due to Sprouty (Sty) expression. In addition to the miRNAs mentioned, several studies have also contributed to an understanding of the role of miR-135 and miR-137 in regulatory control of key active zone proteins [144,145,146]. In a similar way to which microglial extracellular vesicles deliver miRNA to neurons, neurons have also been identified to communicate with glia in the same way [147,148]. Peng, Harvey, Richards and Nixon [147] recently isolated neuron-derived EVs (NDEVs) on microglia from rat cortical neurons and added them to cultured rat microglia, which resulted in improved microglia viability and reduced gene expression of TNF-α, IL-6, MCP-1 and iNOS. Yang et al. [148] further demonstrated that neuronal EV transfer of miR-98 to microglia after ischemic stroke prevented certain neurons from microglial phagocytosis. Future work in this domain should confirm the role of other glia-derived miRNAs essential for synaptic plasticity through ultrastructural data.

In vivo studies have demonstrated a role for microglia in shaping structural synaptic plasticity by phagocytosing the perisynaptic ECM [149,150]. Biswas et al. [151] found that interaction with astrocyte-derived ECM via β1-containing integrin influenced microglial activation. In mice retina, microglia are shown to be hyperactivated while microglial TGF-β1 expression is down-regulated in the absence of γ3-containing laminins [151]. Stoyanov et al. [152] found that chondroitinase ABC (ChABC) treatment promoted microglial recruitment to a lesion site three days following the induction of damage in 5xFAD mice but reduced migration in control animals. Histological analyses have revealed that activated microglia engulf damaged perineuronal nets (PNNs) in the 5xFAD brain, with PNN material identified in both mouse and human microglia [153]. Further, Strackeljan et al. [154] observed that the depletion of microglia increased perisynaptic ECM proteoglycan brevican expression, accompanied by elevated expression of presynaptic marker vGluT1 and increased dendritic spine density. Collectively, these studies suggest an interplay between synaptic elements including microglia process tips, the ECM, and key molecules in glia-neuron communication, in regulating structural synaptic plasticity.

### 3.5. Phagocytic Elimination of Synapses by Microglia Impact Balance between Excitatory and Inhibitory Synapses

In vivo studies over the last decade support the concept that astrocytes contribute to higher brain functions such as learning and memory [155], as well as pathophysiologies such as mood and emotional disorders [156]. Initially, astrocytes were considered to play only less complex roles, such as providing trophic and metabolic support, supporting neuronal survival, differentiation, neuronal guidance, neurite outgrowth and synaptogenesis, while actual brain function would arise exclusively from neuronal activity. Growing evidence has emerged over the last decade to suggest that brain function may arise from neuron–glia interactions.

Microglia create transient contacts with astrocytes, oligodendrocytes as well as neuronal synapses via their motile processes in physiological conditions. Wake, Moorhouse, Jinno, Kohsaka and Nabekura [97] found that the contact time between microglial processes and synapses increases during prolonged ischemia, thereby increasing the likelihood of synaptic elimination. The observation of increased microglial contact time with synapses have been reported in long-term potentiation (LTP). Pfeiffer et al. [157] observed that in mice hippocampal LTP, microglia increased the number of their processes and prolonged their physical contacts with dendritic spines. In LTP, high calcium levels or higher neuronal activity has been demonstrated to result in an increased expression of synaptic AMPA receptors [11,158]. Recently, Saw et al. [159] reported that PI3K in microglia is regulated by epigenetic factors which alter expression of BDNF and thereby downregulate LTP. Sa de Almeida et al. [160] found that Sirt2-deficiency in microglia resulted in LTP impairment in mice hippocampal slices. Hoshino et al. [161] demonstrated that impaired LTP induced by minocycline was alleviated following IL-1 receptor antagonist injection in mice hippocampal sepsis-induced encephalopathy. These findings suggest that prolonged contact time between microglia and synapses may be indicative of long-term synaptic activity.

In persistent pain, glia have been shown to change phenotypes and release chemical mediators resulting in signalling cascades that contribute toward neuronal hypersensitivity. TNF, IL-1β, IFNγ, CCL2 and reactive oxygen species (ROS) have been shown to directly modulate excitatory synaptic transmission by enhancing glutamate release in neuropathic models. Cytokines have been previously shown to be required for normal surface expression of AMPA receptors at synapses. Increased TNF-α concentrations cause a rapid exocytosis of AMPA receptors in hippocampal pyramidal neurons, resulting in increased excitatory synaptic strength [10,11,158,162]. Yu et al. [163] recently identified a microglial P2Y12 receptor-dependent GTP-RhoA/ROCK2 signalling pathway that upregulates excitatory synaptic transmission in the spinal cord dorsal horn, resulting in nociceptive allodynia. Reischer et al. [164] demonstrated that prolonged IFN-γ treatment increases synaptic strength at C-fibre synapses in spinal lamina I rat neurons mediated by microglial activation. Thus, microglial release of chemical mediators may shape synaptic strength. We also note that microglia-derived cytokines can induce neuronal death by phagocytosis [165,166,167], thus influencing the balance between excitatory and inhibitory synaptic transmission.

An imbalance in the ratio of excitatory to inhibitory (E/I) synaptic activity has emerged as a common pathophysiological mechanism in several neuropathological conditions such as Alzheimer’s Disease (AD), autism spectrum disorder (ASD), epilepsy and schizophrenia (SZ). In fact, loss of synaptic connections are considered one of the earliest pathologic hallmarks of AD [168]. Fatemi et al. [169] demonstrated reduced expression of GAD65, GAD67 and receptor subunits GABARA1, GABARA2, GABARA3 and GABARB3 in the parietal cortex and cerebellum of human autism cases compared with controls. Ultrastructural studies of schizophrenia cases have demonstrated an increased density of excitatory synapses in several areas of the basal ganglia and substantia nigra as well as a decreased density of inhibitory synapses [170,171]. Mitew et al. [172] identified a reduction in excitatory vesicular glutamate transporter 1 (VGlut1) boutons in end-stage AD cases and in preclinical AD. In contrast, inhibitory GABAergic synapses were found to be preserved in both human AD and mouse transgenic samples [172]. Recent evidence is suggesting, however, that this may not be the case for all; Kurucu et al. [173] used high-resolution array tomography imaging to demonstrate plaque-associated loss of inhibitory synapses and accumulation of Aβ plaques in a small subset of inhibitory presynaptic terminals. The difference in these findings can be attributed to the methodology used—that is, proteomic studies of synaptic fractions from human AD cases have not been able to yield any changes in inhibitory synaptic proteins. Petrache et al. [174] reported decreased canonical Wnt signalling activity affecting the lateral entorhinal cortex (LEC) with synaptic hyperexcitation, disrupted excitatory–inhibitory inputs onto principal cells, a synaptic imbalance and reduction in the number of parvalbumin-containing (PV) interneurons, and a reduction in the somatic inhibitory axon terminals in the LEC. Petrache et al. [174] also found that targeting GABA_A_ receptors on PV cells restored the identified excitatory–inhibitory imbalance, suggesting a reversible role of excitatory to inhibitory synaptic ratios in determining the progression of AD pathophysiology. Davenport et al. [175] identified a role for CYFIP1 in regulating synapse number and the excitatory–inhibitory synapse ratio in ASD and schizophrenia. More specifically, Davenport, Szulc, Drew, Taylor, Morgan, Higgs, López-Doménech and Kittler [175] found that CYFIP1/CYFIP2 upregulation increased excitatory synapse number while decreasing the size and amplitude of inhibitory synapses and their miniature inhibitory post-synaptic currents (mIPSCs). Furthermore, in vivo knockout in neocortical principal cells increased the expression of post-synaptic GABA_A_ receptor β2/3-subunits and neuroligin 3, thus enhancing synaptic inhibition. Howell and Smith [176] have provided a detailed review of several synaptic structural proteins, such as SHANK, Gephyrin, Homer, PSD-95, and SAPAP, that regulate the excitatory–inhibitory synaptic balance across various brain regions in ASD.

Taken together, abnormal glial function is increasingly recognised as an early pathological feature commonly observed in neurodegenerative and psychiatric diseases. Both astrocytes and microglia have been demonstrated to influence synapse formation and elimination under various physiological conditions [177,178,179,180]. For example, Choudhury, Miyanishi, Takeda, Islam, Matsuoka, Kubo, Matsumoto, Kunieda, Nomoto and Yano [180] found that microglia eliminate assumed weak synapses opsonized with eat-me signals C3 or MFG-E8 during every sleep phase via phagocytosis. In contrast to physiological conditions, glial dysfunction—as observed in pathological contexts—is proposed to result in the disruption of the excitatory/inhibitory synaptic balance [181,182,183]. In colony-stimulating factor 1 receptor (Csf1r) knockout mice, microglia-mediated climbing fiber (CF) elimination was found to be inhibited, thereby attenuating inhibitory GABAergic synaptic transmission [184]. Cantaut-Belarif et al. [185] have shown that microglia differentially control GABAergic and glycinergic inhibition in the spinal cord. Ren et al. [186] found that the microglia-specific Trem2^R47H^ Alzheimer-linked variant increased glutamatergic neuronal transmission and suppressed LTP by enhancing brain TNF-α concentrations in rats. Thus, the role of microglia in influencing the excitatory to inhibitory synaptic ratio appears to present an example where the penta-partite microglial synapse is most relevant in brain pathologies where microglial activation has been described on imaging and/or histologically.

## 4. Conclusions

Over the course of the last decade, microglia have undoubtedly become a force at the CNS synapse that needs to be reckoned with. The concept of a microglial contribution to normal synaptic function should provide a stimulus for additional studies into the detailed tissue distribution of microglial penta-partite synapses, their role in CNS physiology as well as their changes under pathological conditions. Microglial penta-partite synapses represent only a subset of synapses but likely one that has significance for the symptoms reported under conditions of “neuroinflammation”. The latter term is misleading and should be abandoned, but the microglial activation that characterises “neuroinflammation” deserves further scrutiny. Penta-partite aptly describes the type of synapse to look out for, but we agree with the authors of a recent paper on the “active milieu” [187], a new integrating morpho-functional concept of the synapse, that a holistic view is preferrable to an enumeration of synaptic components.

## Figures and Tables

**Figure 1 ijms-22-13186-f001:**
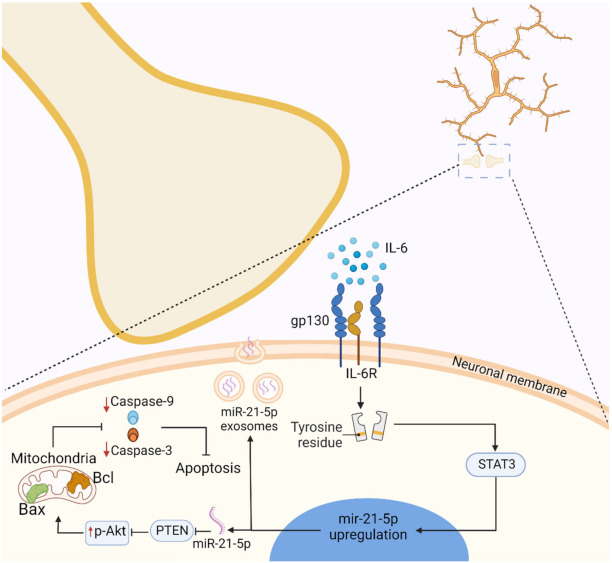
Neuronal gp130 signalling via IL-6R activates STAT3, which causes miR-21-5p upregulation, inhibition of PTEN, and decreased Caspase-3 and -9, thus reducing the likelihood of apoptosis; miR-21-5p upregulation also results in the release of miR-21-5p exosomes.

**Figure 2 ijms-22-13186-f002:**
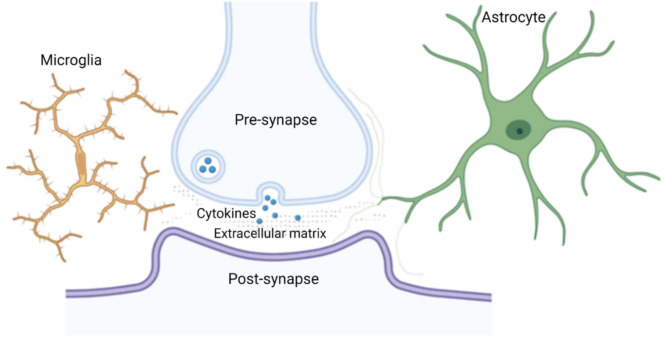
Schematic illustration of the penta-partite synapse consisting of the pre- and post-synaptic neuronal membranes, astrocyte processes (green), extracellular matrix components (dots) and a microglial process (orange) that may be transiently present.

**Table 1 ijms-22-13186-t001:** Cytokines and mRNA transcripts for cytokine receptors identified in unstimulated human, rat and mice microglia and neurons.

Cytokine Family	Cytokines	mRNA Transcripts for Cytokine Receptors	Cell-Type
Interleukins	IL-1β IL-6 IL-8 IL-10 IL-12 IL-15	IL1-RI IL1-RII IL-5R IL-6R IL-8R IL-9R IL-10R IL-12R IL-13R IL-15R gp130	Human microglia [66]
TNF	TNF-α	TNFRI TNFRII	Human microglia [66]
Macrophage Inflammatory Proteins	MIP-1α MIP-1β MCP-1		Human microglia [66]
CXC chemokines	CX3CL1 SDF-1α CXCL12	CX3CR1 CXCR4/CXCR7	Human, mice microglia [67,68] Rat, mice neuron [69,70,71,72,73]
CC chemokines	MCP-1 (CCL2) MIP-1α (CCL3) MIP-1β (CCL4) RANTES (CCL5) C10/MRP-1 (CCL6)		Rat, mice neurons [69,70,71,72,73] and mice microglia [74]

**Table 2 ijms-22-13186-t002:** Common microglial mediators and their receptors involved in synaptic plasticity.

Mediators	Receptor	Receptor Family
ATP, ADP	P2X4R, P2X7R	Purinergic
	P2Y2R, P2Y4R, P2Y6R, P2Y12R, P2Y14R	
	A1R, A2AR, A2BR, A3R	Adenosine
CX3CL1	CX3CR1	CX3 chemokine
CXCL10	CXCR3	CXC chemokine
CXCL12	CXCR4	
CCL2 CCL3 CCL5	CCR2 CCR3 CCR1, CCR3, CCR5	C-C chemokine
Glutamate	NMDAR	Ionotropic glutamate
IL-1β, TNF-α	AMPAR	
GABA	GABA_A_R GABA_B_R	GABA
IL-1β	IL-1R1	Interleukin-1
C3, C1q	CR3	Complement
CD200	CD200R	OX-2 membrane glycoprotein
TGF-β1	TGF-β1R	Transforming growth factor
NA	β2-AR	Adrenoceptor family of the 7-transmembrane superfamily

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
