# Peer review of "Cytokine Signalling at the Microglial Penta-Partite Synapse"

_ijms, 2021, doi:10.3390/ijms222413186_

Round 1
Reviewer 1 Report
This is a comprehensive and detailed review of the role of microglia and cytokines in synaptic transmission under physiological and pathological conditions. The authors of this review show that microglia interact with a subset of synapses in the central nervous system called penta-partie synapses, which are formed by presynaptic and postsynaptic neurons, astrocytes, extracellular matrix and microglia cells. The review is focused on the structure, function, location and role of these synapses, it shows that phagocytic elimination of synapses by microglia impact balance between excitatory and inhibitory synapses which play a role in both in health and disease. Another conclusion is that the term “neuroinflammation”, used usually in relation to activated microglia, is not correct and viable, because activated microglia or the release of their cytokines may occur independent of inflammation and penta-partite synapses are not usually active in a neuroimmunological sense.
Major comments
- Given the well-documented roles of purinergic signaling in neuron–astrocyte communication and since microglia also release ATP and express P2X4 and P2X7 purinergic receptors, I suggest to include a paragraph or chapter about the role of purinergic signaling at the microglia penta/parite synapses.
- This review is focused on cytokine signaling, but the mechanism of release and intracellular mechanisms of cytokine action is explained nowhere in the text.
- The authors should also consider to insert a list of microglia mediators and receptors. For example P2Y12, β2-adrenergic, GABAA and BABAB receptors are mentioned but not explained
Minor comments
Line 39: abbreviation „ECM“ is explained 4 lines above, here it is for the second time
Line 123: abbreviation „TSPO“ and the role of this translocation protein is not explained.
Line 346: abbreviation „P2Y12“ needs explanation
Line 372: abbreviation „ECM“ is explained above
Lines 390-391: Refernces 161 and 162 are missing in the list of References (there are only 158 references)
Line 418, should be: GABARA1, GABARA2, GABARA3 and GABARB3
Line 436: GABA receptors type A or B are not consistently written
Author Response
Please refer to the attachment for the point-by-point response to reviewer 1's comments.

Reviewer 2 Report
The review by Prof. Manuel B. Graeber et al. addresses a crucial topic: microglia's involvement in synaptic transmission. There is currently some misunderstanding of the role of neuroglia in the healthy brain, so this thorough review is timely and valuable for neuroscientists.
I don't have major comments, and the review can be published as presented.
However, it seems to me that this review could have attracted more attention if:
- it had a special section on the role of microglia in long-term synaptic plasticity;
- the primary data on changes in сytokine signaling in different CNS pathologies were summarized in the table
- there is some lack of illustrative material. If the review had some illustrations from the cited articles, it would be very convenient for the readers.
- Schematic illustration of a penta-partite synapse is not very informative.
Minor comments:
Not all abbreviations are explained; for example, TSPO
There are a few typos in the text, for example, line 129 (research interst)
Author Response
Please refer to the attachment for a point-by-point response to Reviewer 2's comments.
